# Detection of Ochratoxin A in Tissues of Wild Boars (*Sus scrofa*) from Southern Italy

**DOI:** 10.3390/toxins17020074

**Published:** 2025-02-06

**Authors:** Sara Damiano, Consiglia Longobardi, Lucia De Marchi, Nadia Piscopo, Valentina Meucci, Alessio Lenzi, Roberto Ciarcia

**Affiliations:** 1Department of Veterinary Medicine and Animal Production, University of Naples Federico II, Via Federico Delpino, 80137 Napoli, Italy; sara.damiano@unina.it (S.D.); consiglia.longobardi@unina.it (C.L.); nadia.piscopo@unina.it (N.P.); rciarcia@unina.it (R.C.); 2Department of Veterinary Sciences, University of Pisa, Viale delle Piagge, 56124 Pisa, Italy; valentina.meucci@unipi.it (V.M.);

**Keywords:** Ochratoxin A, wild boars, liver, kidney, muscle, analytical quantification, bioindicator

## Abstract

Ochratoxin A (OTA) is a secondary metabolite produced by fungi of the genera *Aspergillus* and *Penicillium*, known to contaminate various food substrates. Its toxic effects include direct nephrotoxicity, as well as observed teratogenic, immunogenic, and carcinogenic effects. Climate change may contribute to increased humidity and temperature, favouring fungal growth and, consequently, OTA spreading. Recent increases in wild boar populations, along with their omnivorous nature and their varied diet, define them as environmental bioindicators for contaminants like mycotoxins. This study aimed to assess the concentrations of OTA in kidney, liver, and muscle tissue samples from 74 wild boars that were hunted in different areas of Avellino, Campania region, between 2021 and 2022. Tissue samples underwent extraction, purification, and analysis using high-performance liquid chromatography (HPLC) coupled with a fluorescence detector. Results revealed OTA presence in 35.1% of tested wild boars. The highest OTA concentration was observed in the kidney and liver, with less in the muscle, indicating the presence of this mycotoxin in the wild boars and their surrounding environment. Consequently, there is a need to formulate rules for edible wildlife products. These findings emphasize the significant risk of OTA contamination in wild boar tissues, suggesting their potential as reliable environmental markers for mycotoxin prevalence and as a toxicological concern for human health.

## 1. Introduction

Global climate change is predicted to increase the impact of mycotoxins in temperate countries, including Italy [1]. Among them, Ochratoxin A (OTA), a secondary metabolite of fungi belonging to the *Aspergillus* and *Penicillium* species, has become a subject of growing interest due to its widespread occurrence and significant toxicological impact [2]. In fact, toxigenic fungi belonging to previously mentioned genera can proliferate on dry-meat (especially ham and sausages) surfaces due to their adaptability to low pH and high concentrations of salts [3]. Given the extensive geographical diffusion of OTA, and the significant quantity of potentially contaminated food products resulting from the challenges in its eradication, both animals and humans are inevitably subjected to foods contaminated with this mycotoxin [4]. The risk for human exposure to OTA, both directly through grains consumption and indirectly via the edible tissues of animals, made it worthy of attention from health authorities. Besides its recognized nephrotoxic [5], hepatotoxic [6], and immunotoxic [7] effects, OTA is recognized as carcinogen belonging to group 2B (IARC classification) and, because of its genotoxicity, it can induce chromosomal aberrations in human lymphocytes [8,9]. Unexpectedly, chronic exposure to OTA might be even worse than acute one [10,11]. Small quantities of OTA-poisoned food over time may result in a variety of metabolic, physiologic, and immunologic disorders as well. Consequently, the European Union has set maximum permissible levels for OTA in several food products. Up to date, 1 µg/kg is the limit established by the Italian Ministry of Health for domestic *Sus scrofa*. Exceeding this limit can result in product recalls, thereby impacting marketability and causing financial losses for producers.

Wildlife species, especially those that inhabit agricultural areas or ecosystems tainted by mycotoxins, are susceptible to OTA exposure [12]. The wild boar is one of the most extensively hunted wild ungulates in Italy, occupying a unique ecological function since it traverses extensive geographical regions and adapts to many habitats while using available resources. The wild ungulate population has increased in recent decades mostly due to a lack of natural predators and the depopulation of rural zones [13].

Research on wild boars demonstrates their capacity to retain OTA in their tissues [14,15], indicating environmental contamination and potentially affecting their own and consumers’ health. Consequently, wild boar is regarded as a sentinel species concerning ecological and public health. In fact, the population of wild boars and their feeding behaviour can indicate alterations in ecosystems, including the availability of food resources [16]. In Italy, the wild boar population is widespread and although no official survey has been carried out during the hunting period of this study, the Campania region is among the most densely populated areas. Thus, there has been a rise in wild boar game-meat consumption, especially in the internal area, like the province of Avellino. Although OTA presence is surveyed in most edible animals, wildlife meat surveillance is not included. Given the lack of control over wild animals’ dietary intake and the poorly understood long-term toxic effects of mycotoxin exposure, potential risks to the health of humans require immediate attention, emphasizing the importance of an extensive legislative structure addressing meat contaminants in wildlife.

Monitoring the prevalence of OTA in wild boars serves as an effective method for evaluating environmental pollution, particularly in the regions like Campania characterized by significant anthropogenic influence and interactions between animals and agricultural or forestry practises. Understanding the epidemiological behaviour of this toxic substance is crucial for formulating an effective monitoring and management strategy for public health, whose risk assessment has often been evidenced by EFSA [17].

Many studies in Italy have investigated the prevalence of OTA in specific local products (wine, cereals, cured meats, and hams) [18,19,20], as the Mediterranean climate favours the proliferation of fungus that produce this toxin. Nevertheless, limited studies focus on meat products that come from wildlife. This work aims to explore the prevalence of OTA in the wild boar population of the Campania region. The study also seeks to address how OTA contamination in wildlife could affect ecosystems and pose a risk to food safety, since the detection of OTA residues in animal-derived products, such as meat and organs, raises food safety concerns.

## 2. Results

The analytical method proved to be effective for the accurate quantitative measurement of OTA in various tissues of wild boars. The results of the validation study are presented in Table 1, and the chromatographic representations are shown in Figure 1.

Statistical analysis confirmed that no significant differences were observed in repeatability or reproducibility values across the tested conditions (*p* > 0.05).

The results demonstrated the presence of OTA in 35.1% (n = 26 out of 74) of the analyzed wild boars, with a median value of 0.56 μg/kg (range: 0.03–3.8 μg/kg). OTA was detected in the kidneys of 25 wild boars (33.8%) with an average concentration of 0.69 μg/kg (range: 0.25–2.67 μg/kg), in the liver of 26 wild boars (35.1%) (median: 0.24 μg/kg, range: 0.03–2.13 μg/kg), and in the muscle tissue of 10 wild boars (13.5%) (median: 0.43 μg/kg, range: 0.03–3.8 μg/kg). No significant differences in OTA concentrations were observed among the different type of tissues (*p* > 0.05) (Figure 2).

Of the 26 OTA-positive samples, 16 were from females and 10 from males. OTA concentrations in different tissues by sex are reported in Table 2, showing no significant differences in OTA levels across tissues for each sex, nor between sexes for each tissue (*p* > 0.05). Similar results were found for the age class (17 young and 9 adult age classes), with no significant differences in OTA concentrations across tissues between different age groups (Table 3) (*p* > 0.05).

A positive correlation was observed between OTA levels in the kidney and muscle (moderate correlation), as well as between the kidney and liver (strong correlation), whereas only a weak correlation was found between muscle and liver OTA levels (Table 4). The graphical representation is reported in Figure 3. No significant correlation was detected between OTA concentrations in the tissues and the animals’ age-class or sex.

## 3. Discussion

Studies on wild boars revealed heightened contamination levels of xenobiotics, particularly heavy metals and mycotoxins, in comparison to farmed livestock. Consequently, they are considered sentinels for emerging health issues [21,22,23,24]. Therefore, by investigating OTA exposure in wild boars, we can gain insight into the broader environmental significance of mycotoxin contamination.

In this study, OTA was identified in 35.1% of the wild boars of the Campania Region, more significantly affecting the liver and kidneys than the muscles. The findings align with the metabolic pathways adopted by OTA for excretion and confirm that the kidney is recognized as the target organ for OTA intoxication [25]. The detected level of OTA in the kidney confirms the sensitivity of this species to this mycotoxin, that could also lead to porcine nephropathy [26]. In fact, it is worrying that domestic pigs exposed to a diet containing OTA levels within the recommended limits established by the European Commission (EC 576/2006) experience alteration in the kidneys’ genome [10,27].

Concentrations of OTA have also been identified in the liver of wild boars at levels comparable to those reported in the livers of swine sold in Chinese local supermarkets [28]. This is related to its toxicokinetic: wild boar and pig belong to the same species and, although the eating habits depend on their lifestyle (domestic or wild), OTA metabolism is almost identical [29]. Upon absorption, OTA exhibits strong affinity to plasma albumin and is distributed in the liver, where an important part of its metabolism occurs. The liver is a key organ in the excretion of OTA, while also serving as a temporary reservoir for the toxin. In fact, a part of OTA can be reabsorbed by the intestine via the enterohepatic circle, and this could explain the persistence of OTA in the liver of animals long after exposure [25,30]. This may elucidate why OTA concentrations in this organ are comparable to those in the kidneys.

Since a reduced number of muscle samples resulted positive to OTA contamination, a smaller risk for meat consumers may be expected. However, wild boar meat is used to produce niche products, as coppa and salami, and may contribute to enrich OTA concentration because of the loss of water during drying stage [31]. Consequently, OTA carry-over to humans and, above all, children, is even more hazardous [32]. One main problem is that meat is also vulnerable while ripening since moulds commonly observed on exposed muscle surfaces may produce OTA [33]. In a study conducted in Veneto, analyzed artisan salami were found to contain OTA amounts within the Italian guideline values (1 µg/Kg). However, one of them contained a concentration that exceeded the legal limits [34]. This highlights the dangers of consuming wild boar-derived secondary products (especially for children), strengthening the hypothesis of the present study, i.e., products derived from wild boar meat are a possible source of OTA. In fact, it cannot be ruled out that the habitual intake of specific regional specialities for self-consumption, not yet integrated into the exposure assessment, may contribute to human OTA exposure [35].

The current study indicates that OTA levels are unaffected by sex or age-class, according to OTA analysis on tissues of wild boar belonging to northern Italy [15]. Therefore, monitoring could be simplified: randomized analysis may prove effective. At the same time, the consistency in contamination levels highlighted the necessity for detailed ecological assessments to determine the primary exposure to OTA in the sampling area.

Considering the importance of wild boars as prey within the broader food chain, there may be consequences for predators who consume them, potentially resulting in OTA exposure throughout several trophic levels. Consequently, examining OTA levels in additional wildlife species, particularly those cohabiting with wild boars, might provide a more comprehensive understanding of their habitat.

These results highlight a regulatory weakness and emphasize the necessity for awareness among hunters and wildlife consumers concerning OTA contamination risks.

Since OTA contamination is present at various trophic levels, a variety of techniques to alleviate its toxicity have been explored [36]. For example, proper agriculture and storage techniques can help to avoid contamination of cereals, which are frequently used in wild-animal diets. In addition, this type of research can spark a larger debate about the need of adopting specialized methods for the control of wild game meat, which is becoming increasingly important in the human diet due to its beneficial content.

Although OTA is one of the most prominent mycotoxins, it is infrequently detected in isolation inside animal tissues or food products [37]. In fact, recent studies have identified Zearalenone (ZEA) in the kidney, liver, and muscles of wild boars belonging to the Campania region [38]. The concurrent presence of many mycotoxins may exert a synergistic effect, amplifying total toxicity. Consequently, a precise risk assessment should include the monitoring of co-contamination.

## 4. Conclusions

Biomonitoring using wild boars enables the identification of particularly vulnerable areas, notably agricultural areas, where mycotoxin contamination poses risks to both animals and the supply of food for humans. Although further investigations are required, this work provides an important basis to understand the distribution of OTA in wild boars within the province of Avellino, an area characterized by a strong integration between wild environments and human activities.

The study enhances awareness within the scientific community, regulatory agencies, and consumers about the necessity of incorporating biomonitoring into land management policy. This collaborative approach may enhance the formulation of prevention strategies, mitigating public health concerns and safeguarding the environmental integrity of rural and forested regions. The utilization of wild boars as biological indicators serves not only as a research tool but also as a catalyst for enhancing awareness of the hidden dangers to the sustainability of agroforestry ecosystems.

## 5. Materials and Methods

### 5.1. Samples Collection and Ethical Approval

For this study, a convenience sampling was carried out. The samples were taken from 74 wild boars from the province of Avellino, Italy. The animals were legally killed in their natural environment by licenced hunters in accordance with the 2021–2022 annual hunting plan sanctioned by the province of Avellino, Campania region. Therefore, no ethics committee permission was necessary as the wild boars were not slaughtered for research purposes. For each animal, a datasheet was compiled, recording its weight, sex, and age class (young or adult). In total, 38 were female, and 36 were male, with weights varying from 30 to 140 kg. The age was determined using the tooth eruption method [39] and varied from 6 months to 2 years for young animals, and older than 2 years for adults. Samples of the liver, muscle (biceps femoris), and the kidneys were freshly collected from each wild boar, individually packaged in sterile tubes, and stored at −20 °C until laboratory investigations were conducted. The laboratory analysis to assess OTA levels were carried out at the Pharmacology and Toxicology Laboratory of the Department of Veterinary Sciences, University of Pisa. Figure 4 illustrates the hunting areas involved in this study.

### 5.2. Reagents

Sigma© (Milan, Italy) provided reference standards for OTA (from *Aspergillus ochraceus*) (10 mg/mL in acetonitrile) and ochratoxin B (10 mg/mL). Working solutions were obtained by diluting the stock solutions with a mobile phase made up of methanol and sodium phosphate buffer (pH 7.5) in a 60:40% *v*/*v* ratio. All reagents were sourced from Sigma (Milan, Italy). HPLC-grade water, methanol, ethyl acetate, and acetonitrile were procured from VWR© (Milan, Italy).

### 5.3. Chromatographic Method

The chromatographic system used included a PerkinElmer Series 200 binary pump (Waltham, MA, USA) and a Jasco FT-1520 fluorescence detector (Jasco, Tokyo, Japan). The detector was set with an excitation wavelength of 380 nm and an emission wavelength of 420 nm. Data processing was carried out using Totalchrom Navigator^®^ software 6.3.2. A reversed-phase C18 HAISIL HL column (4.6 mm × 50 mm, 5.0 µm; Higgins Analytical, USA) connected to a WatersGuard-Pak™ C18 pre-column (4 mm) (Waters, Milford, MA, USA) was maintained at 25 °C.

#### 5.3.1. Sample Preparation and OTA Quantification

The sample preparation was described in detail in Iemmi et al. [15] and Monaci et al. [40]. Briefly, samples of the muscle (5 g), liver (5 g or less if unavailable), and kidney (1 g) were collected from each wild boar and homogenized with 5 mL of 1 M phosphoric acid using an Ultra Turrax T25 (IKA, Staufen im Breisgau, Germany) homogenizer for several minutes. An internal standard, OTB (100 µL, 100 ng/mL), was added to the homogenate. The mixture was transferred to a centrifuge tube and extracted with 5 mL of ethyl acetate, vortexed, shaken, and centrifuged at 3000 rpm for 10 min. The organic phase was collected, and the residue was re-extracted using the same procedure, combining the organic phases. The combined organic phase was reduced to approximately 5 mL and back-extracted with 5 mL of NaHCO_3_ (pH 8.4), vortexed, and centrifuged at 3000 rpm for 10 min. The aqueous extract was then acidified to pH 2.5 using 85% H_3_PO_4_ and briefly sonicated to remove CO_2_. Finally, OTA was re-extracted into 5 mL of ethyl acetate, vortexed, centrifuged at 3000 rpm for 10 min, and the organic phase was evaporated to dryness under a nitrogen stream. The residue was reconstituted in 500 µL of mobile phase, consisting of a methanol-phosphate-buffer solution at pH 7.5 (0.03 M Na_2_HPO_4_, 0.007 M NaH_2_PO_4_) in a 40/60% *v*/*v* ratio. A 100 µL aliquot was then injected for HPLC analysis with a flow rate of 1 mL/min.

#### 5.3.2. Method Validation

The HPLC-FLD method was validated in accordance with EU criteria for confirmatory methods for contaminants [14], assessing specificity, recovery, linearity, LOD, LOQ, repeatability, and reproducibility. The Italian Ministry of Health established a limit of 1 µg/kg OTA in pork meat and its derivatives in 1999 [41]. The validation process took this limit into consideration. Linearity was tested by spiking muscle, liver, and kidney samples with OTA at concentrations of 0.25, 0.5, 1.0, 2.5, 5.0, and 7 µg/kg, followed by analysis using the extraction method. This experiment was repeated three times. Repeatability was evaluated by analyzing muscle, liver, and kidney samples spiked with OTA at levels of 0.25, 2.5, and 7 µg/kg, with each sample analyzed in triplicate on the same day. Each contamination level was evaluated three times over a five-day period to ensure within-laboratory reproducibility. These experiments also provided data for determining recovery. The LOD and LOQ were determined using the signal-to-noise approach, with the LOD and LOQ corresponding to signal-to-noise ratios of 3 and 10, respectively. The analytical response and chromatographic noise were measured from the chromatogram of a blank sample extract (1 mL) spiked with an OTA solution.

### 5.4. Statistical Analysis

Repeatability and reproducibility data were statistically analyzed using a one-way ANOVA followed by Tukey’s multiple comparisons test to identify significant differences. Repeatability was assessed by analyzing muscle, liver, and kidney samples spiked with OTA at concentrations of 0.25, 2.5, and 7 µg/kg, with each sample analyzed in triplicate on the same day. Reproducibility within the laboratory was evaluated by testing each contamination level in triplicate over a five-day period.

Data normality was assessed with the Kolmogorov–Smirnov test. Since the data did not follow a normal distribution, results are presented as median and range. A one-way ANOVA, followed by Tukey’s multiple comparisons test, was used to evaluate significant differences among groups based on tissue type, sex, and age-class.

Linear regression and Spearman’s correlation coefficient investigations were performed to evaluate the associations between OTA concentrations in the muscle, liver, and kidney tissues, as well as the age-classes and sex of the wild boars. Correlation strength was classified as strong (r ≥ 0.8), moderately strong (0.6 ≤ r < 0.8), moderate (0.3 ≤ r ≤ 0.5), and weak (r ≤ 0.2). Statistical significance was set at *p* < 0.05.

## Figures and Tables

**Figure 1 toxins-17-00074-f001:**
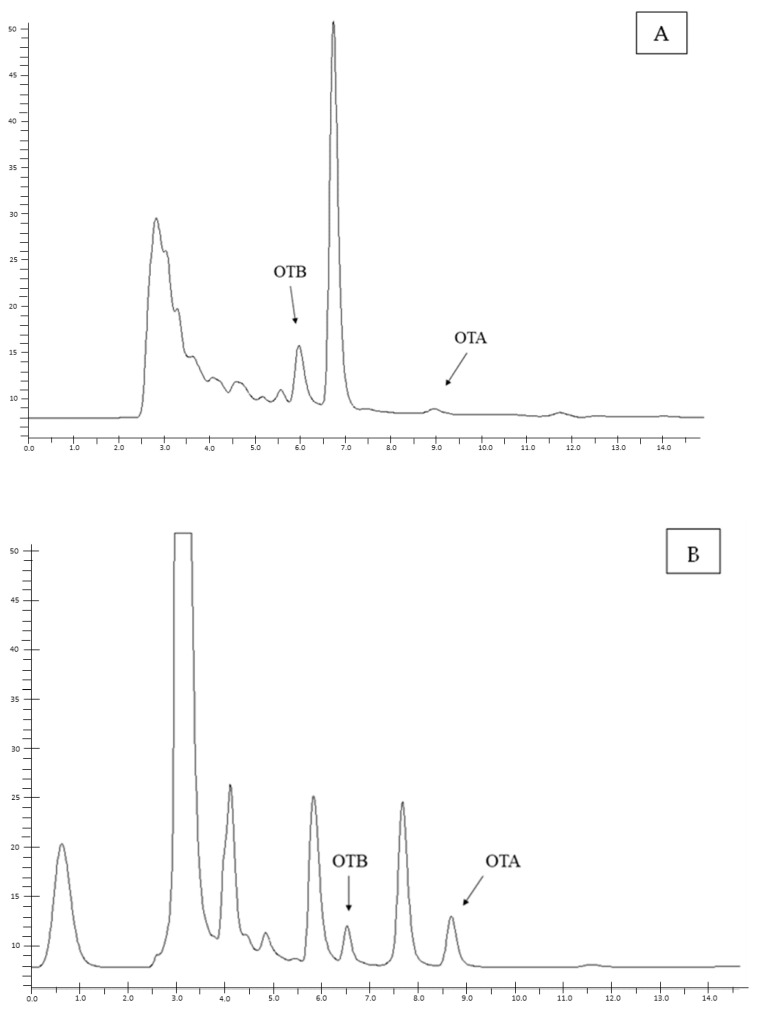
HPLC-FLD chromatograms of: (**A**) a liver sample naturally contaminated with OTA; (**B**) a kidney sample naturally contaminated with OTA; and (**C**) a muscle sample naturally contaminated with OTA, and (**D**) OTA and OTB standard solutions (at 10 ng/mL).

**Figure 2 toxins-17-00074-f002:**
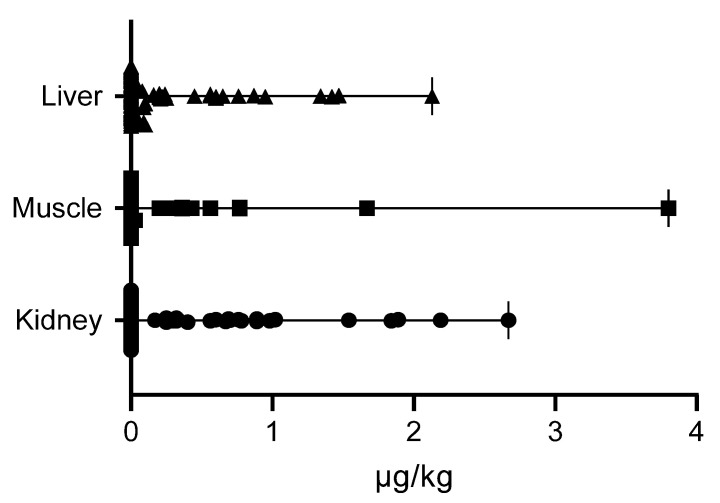
Median ± range of OTA concentrations (μg/kg) in liver, muscle, and kidneys of wild boars.

**Figure 3 toxins-17-00074-f003:**
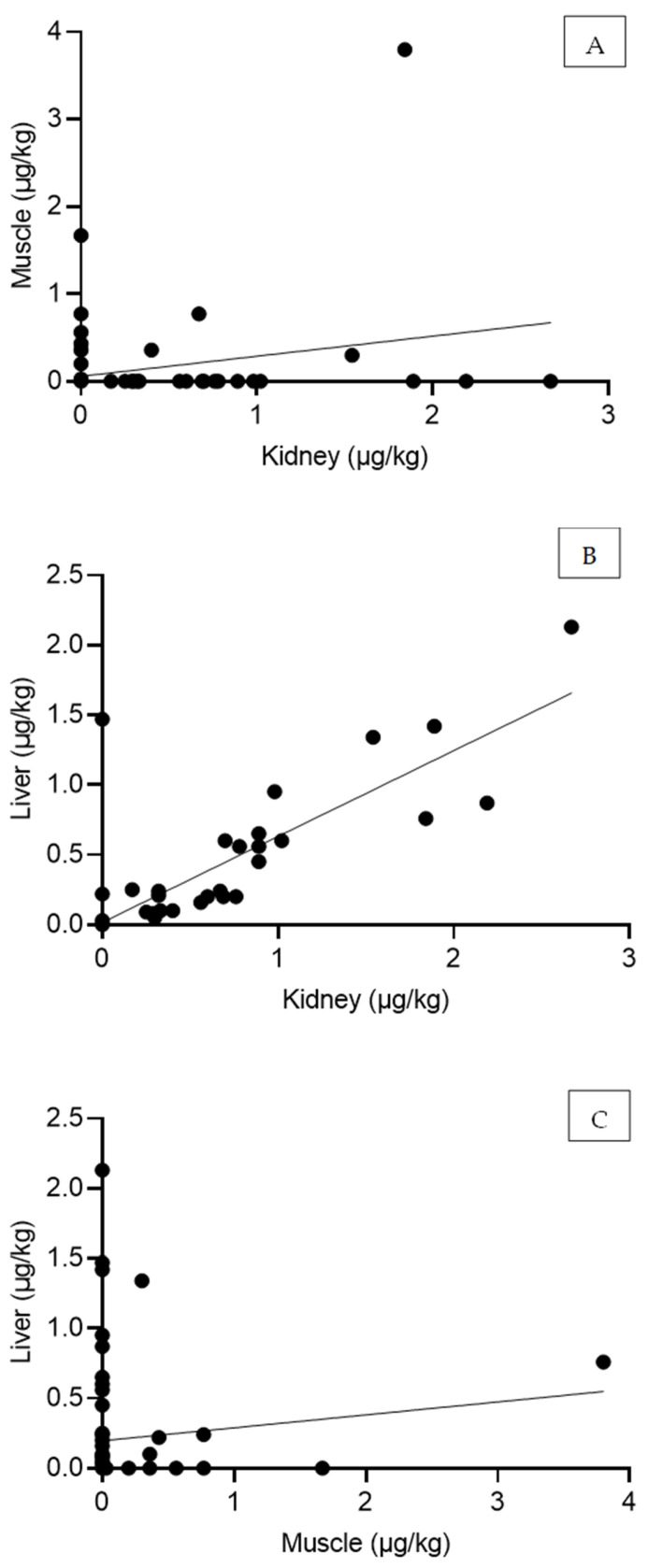
Linear scatterplots depicting the Spearman correlation analysis of OTA concentrations between kidney and muscle (**A**), kidney and liver (**B**), and muscle and liver (**C**) tissues.

**Figure 4 toxins-17-00074-f004:**
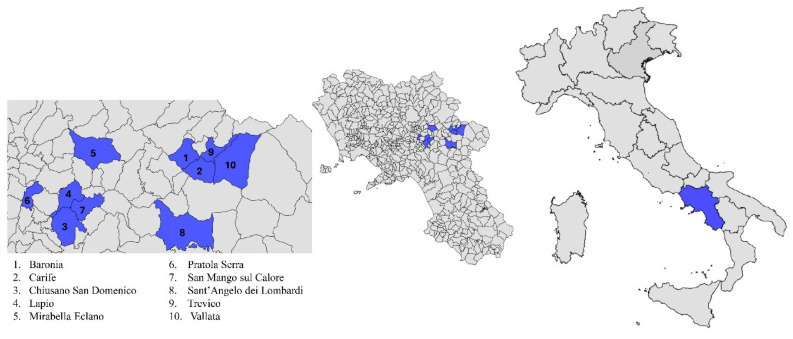
Cartographic representation of the province of Avellino (Campania region, southern Italy) delineating hunting zones (in blue) designated for wild boar (*Sus scrofa*) sampling. The image was created with Epi Info version 7.2.

**Table 1 toxins-17-00074-t001:** Validation criteria for HPLC method; (LOD = limit of detection, LOQ = limit of quantification, r^2^ = coefficient of correlation, SD = standard deviation, RSD = relative standard deviation).

Parameters		Muscle	Liver	Kidney
		OTA		
LOD (μg/kg)		0.01	0.01	0.01
LOQ (μg/kg)		0.02	0.02	0.02
r^2^		0.998	0.996	0.995
Repeatability (µg/kg)				
0.25	Mean concentration ± SD RSD (%)	0.23 ± 0.01 6.20	0.22 ± 0.01 3.64	0.22 ± 0.016.15
2.50	Mean concentration ± SD RSD (%)	2.50 ± 0.25 10.00	2.32 ± 0.27 11.89	2.21 ± 0.209.08
10.0	Mean concentration ± SD RSD (%)	10.07 ± 0.30 3.03	9.73 ± 0.70 7.22	9.45 ± 0.222.30
Reproducibility (µg/kg)				
0.25	Mean concentration ± SD RSD (%)	0.23 ± 0.02 8.02	0.22 ± 0.02 8.65	0.21 ± 0.016.17
2.50	Mean concentration ± SD RSD (%)	2.29 ± 0.30 13.37	2.41 ± 0.31 4.03	2.11 ± 0.209.48
10.0	Mean concentration ± SD RSD (%)	9.74 ± 0.33 3.45	9.52 ± 0.43 4.48	9.50 ± 0.232.42
Recovery (%)				
0.25		91.57 ± 8.01	90.29 ± 7.67	87.86 ± 4.10
2.5		94.14 ± 12.64	97.86 ± 15.04	83.86 ± 7.71
10.0		98.17 ± 3.37	95.80 ± 4.38	94.43 ± 2.22

**Table 2 toxins-17-00074-t002:** Ochratoxin A (OTA) concentrations (μg/kg) in liver, muscle, and kidneys of wild boars according to gender. All data are presented as median and range.

Sex	*N*	Kidney	Muscle	Liver
Male	10	0.65 (0.32–2.67)	0.43 (0.08–3.80)	0.21 (0.10–2.13)
Female	16	0.68 (0.17–2.19)	0.56 (0.03–3.80)	0.25 (0.03–1.47)

**Table 3 toxins-17-00074-t003:** Ochratoxin A (OTA) concentrations (μg/kg) in liver, muscle, and kidneys of wild boars according to age class. All data are expressed as median and range.

Age Class	*N*	Kidney	Muscle	Liver
Young	17	0.65 (0.25–2.67)	0.77 (0.20–3.80)	0.22 (0.03–2.13)
Adult	9	0.67 (0.17–1.89)	0.25 (0.03–0.56)	0.24 (0.08–1.42)

**Table 4 toxins-17-00074-t004:** Data of Spearman correlation analysis between muscle, liver, and kidney OTA concentrations. Strong (r ≥ 0.8), moderately strong (0.6 ≤ r < 0.8), moderate (0.2 ≤ r ≤ 0.5), and weak (r ≤ 0.2).

	*r*	*p*
Kidney vs. Muscle	0.26 (0.025–0.46)	0.03
Kidney vs. Liver	0.84 (0.75–0.89)	2.90 × 10⁻²⁰
Muscle vs. Liver	0.11 (-0.12–0.33)	0.34

## Data Availability

The original contributions presented in this study are included in the article. Further inquiries can be directed to the corresponding author.

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
