# Peer review of "Detection of Ochratoxin A in Tissues of Wild Boars (*Sus scrofa*) from Southern Italy"

_toxins, 2025, doi:10.3390/toxins17020074_

Round 1
Reviewer 1 Report
Comments and Suggestions for Authors
Line 14 needs to be reformulated “ 74 wild boars hunted”
Line 20 also needs to be reformulated
Line 76 The conclusions of current studies?
Perhaps there should also be a graphical representation that highlights the discovered correlations (e.g. scatterplots or heatmaps)
Is the number of samples relevant in relation to the existing population in the area? Please provide an estimate of the number of animals in the area.
Author Response
Comments and Suggestions for Authors
- Line 14 needs to be reformulated “74 wild boars hunted”; Line 20 also needs to be reformulated
R: Dear reviewer, we have reformulated the lines you indicated. Thank you.
- Line 76 The conclusions of current studies?
R: Dear reviewer, we ha reformulated. Thank you
- Perhaps there should also be a graphical representation that highlights the discovered correlations (e.g. scatterplots or heatmaps)
R: The authors are grateful to the reviewer for providing the opportunity to enhance the scientific quality of the manuscript. A heatmap has been added (Figure 2) to visually represent the Spearman correlation analyses previously reported in Table 4.
- Is the number of samples relevant in relation to the existing population in the area? Please provide an estimate of the number of animals in the area.
R: Dear reviewer, this epidemiological study utilised convenience sampling. Wildlife research widely uses such sampling for practical and logistical reasons. In fact, access to wild animals can be limited and depends on the hunting season and collaboration with hunters. However, there has never been an official census of wild boars in the province of Avellino.

Reviewer 2 Report
Comments and Suggestions for Authors
Title: Detection of Ochratoxin A in tissues of Wild Boars (Sus scrofa) from Southern Italy
Manuscript Number: toxins-3452771
The paper is on toxins in wild boars. The paper is scientifically sound but some uncompleteness is observed. If some more data should be presented may improve the paper. Some common mistakes are there which are presented below:
Line 48: Ministry of Health, specify the country like Italy
Line 56: wild pigs; use one word either: boars or pig in whole manuscript
Table 1: Anova is missing
Units of Repetability and Reproducibility is missing
Figure 1.: Retention time is different in all the graphs of OTA and OTB
Discussion part is very short. A significant amount of discussion is require in support of your results. Similar for conclusion, add importance of your work and your outcomes.
Line 203: Avellino, Itlay.
Line 211: How samples were collected
Line 224: Information of sodium phosphate buffer purchase is missing/
Line 234: room temperature is missing; Guard coloum details are missing
Line 279: You write that you used anova but in tables it is not observed.
Work on statistical analysis of data.
Author Response
Comments and Suggestions for Authors
Title: Detection of Ochratoxin A in tissues of Wild Boars (Sus scrofa) from Southern Italy
Manuscript Number: toxins-3452771
The paper is on toxins in wild boars. The paper is scientifically sound but some incompleteness is observed. If some more data should be presented may improve the paper. Some common mistakes are there which are presented below:
- Line 48: Ministry of Health, specify the country like Italy
R: Dear reviewer, we have specified the information in the text. Thank you.
- Line 56: wild pigs; use one word either: boars or pig in whole manuscript
R: Dear reviewer, we have homologated the words for the entire manuscript, as suggested. Thank you.
- Table 1: Anova is missing
R: The authors thank the reviewer for their valuable suggestions. To further clarify the statistical analyses, ANOVA results have been added to Table 1, and p-values have been included in the corresponding Results section, as recommended. Additional information was also reported in the material and methods section.
- Units of Repeatability and Reproducibilityis missing
R: Added. Thank you.
- Figure 1.: Retention time is different in all the graphs of OTA and OTB
R: The authors are grateful to the reviewer for the comment, which allows us to clarify this analytical point. While the OTA and OTB peaks in biological matrices exhibited the same/similar retention time, the standards of OTA and OTB—having fewer background interferences compared to biological matrices and being used at higher concentrations (10 mg/mL) than naturally contaminated samples—showed a slight difference in retention time.
- Discussion part is very short. A significant amount of discussion is required in support of your results. Similar for conclusion, add importance of your work and your outcomes.
R: Dear reviewer, thank you for your suggestion. We have implemented the discussion section and conclusions as well.
- Line 203: Avellino, Itlay.
R: We have added the country, i.e., Italy. Thank you.
- Line 211: How samples were collected
R: We have added further information in the sampling paragraph. Thank you.
- Line 224: Information of sodium phosphate buffer purchase is missing
R: The required information has been added. Thank you.
- Line 234: room temperature is missing; Guard column details are missing
R: The required information has been added in the section 5.3 Chromatographic method. Thank you for your suggestion.
- Line 279: You write that you used anova but in tables it is not observed.
R: The authors thank the reviewer for the comment. The statistics are not reported in the tables because, in all cases, the results were not statistically significant, with p-values > 0.05. However, to clarify the results, p-values have been added in the text referring to the tables.
- Work on statistical analysis of data.
R: Done. Thank you.

Reviewer 3 Report
Comments and Suggestions for Authors
This manuscript determined ochratoxin A (OTA) in the tissues of in wild boars from southern Italy. The results underscore the substantial risk of OTA contamination in wild boar tissues, highlighting their potential as effective environmental indicators for mycotoxins prevalence. The manuscript is interesting and easy to read. However, some work is required before the manuscript can be considered for publication. The comments are described below:
1. In the introduction, the authors should present the development of OTA detection in various products, particularly meat products. The distribution of OTA in these products is unclear and requires further elucidation.
2. The conclusion should be rewritten.
3. The reference style should follow the request of Toxins.
Author Response
Comments and Suggestions for Authors
This manuscript determined ochratoxin A (OTA) in the tissues of in wild boars from southern Italy. The results underscore the substantial risk of OTA contamination in wild boar tissues, highlighting their potential as effective environmental indicators for mycotoxins prevalence. The manuscript is interesting and easy to read. However, some work is required before the manuscript can be considered for publication. The comments are described below:
- In the introduction, the authors should present the development of OTA detection in various products, particularly meat products. The distribution of OTA in these products is unclear and requires further elucidation.
R: Dear reviewer, we have added some further information in the introduction (line 42-44). Thank you for your suggestion.
- The conclusion should be rewritten.
R: Dear reviewer, we have modified them. Thank you.
- The reference style should follow the request of Toxins.
R: Dear reviewer, we have modified the reference style accordingly to journal guidelines. Thank you.

Reviewer 4 Report
Comments and Suggestions for Authors
1. The paper titled “Detection of Ochratoxin A in tissues of Wild Boars (Sus scrofa) from Southern Italy” aimed to assess ochratoxin concentrations in kidney, liver, and muscle tissue samples from 74 wild boars hunted in different areas of Avellino, Campania region, from 2021 to 2022. The study has potential but need following changes if considered for publication
2. Title is good and elaborative
3. Abstract is written good but too much background as well as study lacks methodology and potential application of study results so add methodology and elaborate results along with study recommendations
- Introduction is written good but add relevant information for Ochratoxin potential economic loss in Italy as reported by some economic surveys to establish the worth of study
- In Introduction, what about the compliance limit for the reported toxin standard and prevalence in various foods need to be discussed
- Methodology section needs clarity especially for detail method for HPLC analysis conditions
- Results are described in detail but need to be supported with clear understanding of significance and non-significance
- Conclusion need to be elaborative with results and add concrete recommendations for pragmatic application
- Grammatical mistakes observed on several places so there is need to go through the paper for language and grammatical mistakes check
Author Response
Comments and Suggestions for Authors
The paper titled “Detection of Ochratoxin A in tissues of Wild Boars (Sus scrofa) from Southern Italy” aimed to assess ochratoxin concentrations in kidney, liver, and muscle tissue samples from 74 wild boars hunted in different areas of Avellino, Campania region, from 2021 to 2022. The study has potential but need following changes if considered for publication
Title is good and elaborative
- Abstract is written good but too much background as well as study lacks methodology and potential application of study results so add methodology and elaborate results along with study recommendations
R: Dear reviewer, thank you for your valuable comment. We have reformulated the Abstract accordingly to your suggestions.
- Introduction is written good but add relevant information for Ochratoxin potential economic loss in Italy as reported by some economic surveys to establish the worth of study
R: Dear reviewer, thank you for your advice. We have added something to highlight the topic (line 58-59, 84, and 90-92).
- In Introduction, what about the compliance limit for the reported toxin standard and prevalence in various foods need to be discussed
R: Dear reviewer, we have added some more information. Thank you for your suggestion.
- Methodology section needs clarity especially for detail method for HPLC analysis conditions
R: The entire Section 5.3, "Chromatographic Method," has been rewritten to ensure clarity of the methodology employed, as suggested by the reviewer. Thank you.
- Results are described in detail but need to be supported with clear understanding of significance and non-significance
R: The authors thank the reviewer. The Results section has been improved by adding p-values in each section, including an additional statistical analysis regarding the method validation data, and incorporating a graphical representation (heatmaps) to better illustrate the correlation results. The authors hope that these improvements, as suggested also by the previous reviewers, make the results scientifically clearer.
- Conclusions need to be elaborative with results and add concrete recommendations for pragmatic application
R: Dear reviewer, we have modified the conclusions. Thank you.
- Grammatical mistakes observed on several places so there is need to go through the paper for language and grammatical mistakes check
R: Dear reviewer, thank you for your careful reading. We have checked the grammatical mistakes and corrected them.

Round 2
Reviewer 1 Report
Comments and Suggestions for Authors
The authors improved they work in the article “Detection of Ochratoxin A in tissues of Wild Boars (Sus scrofa) 2 from Southern Italy”, but there are still some minor revisions that needs to be done.
The authors have modified part of the paper according to the suggestions, but two concerns remain on my part.
1.
- Perhaps there should also be a graphical representation that highlights the discovered correlations (e.g. scatterplots or heatmaps)
R: The authors are grateful to the reviewer for providing the opportunity to enhance the scientific quality of the manuscript. A heatmap has been added (Figure 2) to visually represent the Spearman correlation analyses previously reported in Table 4.
Do you think it is possible to find another way to scatter those data? Maybe my first suggestion was not understood. A graph showing the correlations but also the values, something like the graphs resulting from PCA.
2.
Is the number of samples relevant in relation to the existing population in the area? Please provide an estimate of the number of animals in the area.
R: Dear reviewer, this epidemiological study utilized convenience sampling. Wildlife research widely uses such samplings for practical and logistical reasons. In fact, access to wild animals can be limited and depends on the hunting season and collaboration with hunters. However, there has never been an official census of wild boars in the province of Avellino.
Maybe that information should be mentioned in the text. You say that there is a large population in the area, on what bases? Maybe a statistic on Campania region, or just mention that there are no data regarding the size of the population in the area.
Author Response
ANSWERS TO REVIEWER 1
Comments and Suggestions for Authors
The authors improved they work in the article “Detection of Ochratoxin A in tissues of Wild Boars (Sus scrofa) 2 from Southern Italy”, but there are still some minor revisions that needs to be done.
The authors have modified part of the paper according to the suggestions, but two concerns remain on my part.
- Perhaps there should also be a graphical representation that highlights the discovered correlations (e.g. scatterplots or heatmaps)
Do you think it is possible to find another way to scatter those data? Maybe my first suggestion was not understood. A graph showing the correlations but also the values, something like the graphs resulting from PCA.
R: The authors have modified the graph again, removing the heatmap and adding linear scatterplots that depict not only the correlation but also the OTA values among the different tissues (Figure 3). Additionally, a graphical representation with medians and ranges of the OTA concentrations found in the three tissues has been added (Figure 2), hoping to have met the reviewer’s requests.
- Is the number of samples relevant in relation to the existing population in the area? Please provide an estimate of the number of animals in the area.
R: Dear reviewer, this epidemiological study utilized convenience sampling. Wildlife research widely uses such samplings for practical and logistical reasons. In fact, access to wild animals can be limited and depends on the hunting season and collaboration with hunters. However, there has never been an official census of wild boars in the province of Avellino.
Maybe that information should be mentioned in the text. You say that there is a large population in the area, on what bases? Maybe a statistic on Campania region or just mention that there are no data regarding the size of the population in the area.
R: Dear reviewer, thank you for your suggestions. We have evidenced the type of sampling in the text in the sample collection section, and we have added the information you suggested as well (line 64-67).

Reviewer 2 Report
Comments and Suggestions for Authors
THE MANUSCRIPT IS NOW LOOKING IN GOOD ORDER AND MOST OF THE CORRECTIONS MADE WELL
Author Response
ANSWERS TO REVIEWER 2
Comments and Suggestions for Authors
THE MANUSCRIPT IS NOW LOOKING IN GOOD ORDER AND MOST OF THE CORRECTIONS MADE WELL
R: Dear reviewer, thank you for helping to improve the quality of the manuscript.

Reviewer 3 Report
Comments and Suggestions for Authors
The manuscript has been improved significantly, and can be considered for publication.
Author Response
ANSWERS TO REVIEWER 3
Comments and Suggestions for Authors
The manuscript has been improved significantly and can be considered for publication.
R: Dear reviewer, thank you for helping to improve the quality of the manuscript.

Reviewer 4 Report
Comments and Suggestions for Authors
The suggested changes by me are incorporated and paper may be considered for publication
Author Response
ANSWERS TO REVIEWER 4
Comments and Suggestions for Authors
The suggested changes by me are incorporated and paper may be considered for publication
R: Dear reviewer, thank you for helping to improve the quality of the manuscript.

Round 3
Reviewer 1 Report
Comments and Suggestions for Authors
The authors improved they work in the article “Detection of Ochratoxin A in tissues of Wild Boars (Sus scrofa) 2 from Southern Italy”. In my opinion the manuscript could be accepted for publication. The representation that i asked for are ok and suggestive.